# Liquid Biopsy to Detect Minimal Residual Disease: Methodology and Impact

**DOI:** 10.3390/cancers13215364

**Published:** 2021-10-26

**Authors:** Natasha Honoré, Rachel Galot, Cédric van Marcke, Nisha Limaye, Jean-Pascal Machiels

**Affiliations:** 1Institute for Experimental and Clinical Research (IREC, Pôle MIRO), Université Catholique de Louvain (UCLouvain) ,1200 Brussels, Belgium; Rachel.galot@uclouvain.be (R.G.); cedric.vanmarcke@uclouvain.be (C.v.M.); 2Department of Medical Oncology, Institut Roi Albert II, Cliniques Universitaires Saint-Luc, 1200 Brussels, Belgium; 3Genetics of Autoimmune Diseases and Cancer, de Duve Institute, Université Catholique de Louvain (UCLouvain), 1200 Brussels, Belgium; nisha.limaye@uclouvain.be

**Keywords:** liquid biopsy, minimal residual disease, ctDNA

## Abstract

**Simple Summary:**

The field of liquid biopsy is rapidly evolving. Techniques that improve accuracy are constantly being developed, and clinicians increasingly use liquid biopsy as a tool to guide their clinical practice. The assessment of minimal or microscopic residual disease (MRD) after oncological treatment with curative intent, however, remains challenging. Therefore, the implementation of liquid biopsy to determine the presence of MRD is not yet standardized. In this review, we focus on the detection of MRD through liquid biopsy in solid cancers, highlighting currently available methodologies and ongoing challenges.

**Abstract:**

One reason why some patients experience recurrent disease after a curative-intent treatment might be the persistence of residual tumor cells, called minimal residual disease (MRD). MRD cannot be identified by standard radiological exams or clinical evaluation. Tumor-specific alterations found in the blood indirectly diagnose the presence of MRD. Liquid biopsies thus have the potential to detect MRD, allowing, among other things, the detection of circulating tumor DNA (ctDNA), circulating tumor cells (CTC), or tumor-specific microRNA. Although liquid biopsy is increasingly studied, several technical issues still limit its clinical applicability: low sensitivity, poor standardization or reproducibility, and lack of randomized trials demonstrating its clinical benefit. Being able to detect MRD could give clinicians a more comprehensive view of the risk of relapse of their patients and could select patients requiring treatment escalation with the goal of improving cancer survival. In this review, we are discussing the different methodologies used and investigated to detect MRD in solid cancers, their respective potentials and issues, and the clinical impacts that MRD detection will have on the management of cancer patients.

## 1. Introduction

### 1.1. Minimal Residual Disease

Minimal or microscopic residual disease (MRD) is defined as ‘a very small number of cancer cells that remain in the body during or after treatment’ [1]. Tumor burden varies with disease and treatment. When there is a response to treatment, tumor burden decreases, and the tumor eventually becomes undetectable in imaging or clinical examination. However, if tumor cells remain in the patient’s body after treatment, they may induce relapse, either locally or through distant metastases (Figure 1). The importance of MRD detection was first highlighted in hematological cancers and allowed clinicians to evaluate the success of therapy and to predict short- and long-term relapse [2]. In solid tumors, the detection of MRD has enabled patients to be classified as having a high or low risk of relapse [3]. Hence, several research strategies are now being investigated to accurately detect MRD in solid tumors and integrate it into treatment strategies.

The ability to detect MRD may prove beneficial for patients and clinicians. After completion of standard therapy with curative intent, treatment could theoretically be adapted based on liquid biopsy results. Treatment intensification in MRD-positive patients has the potential to improve disease-free survival and overall survival if MRD is efficiently treated. At the opposite end, de-intensification strategies omitting adjuvant treatment (for example, systemic chemotherapy) in MRD-negative patients could reduce patient burden, decrease treatment related side effects, improve the quality-of-life of the patient, and reduce financial costs for society without impairing survival rates.

New clinical trials are now being designed or initiated on diverse tumor types in order to establish the role of MRD detection in the selection of the optimal treatment after curative local treatment [4,5,6]. This strategy is limited by the reliable detection of MRD, as it cannot be assessed by standard imagery or clinical examination. Liquid biopsies have become a possible means to overcome this limitation. Although no consensus has yet been reached on the appropriate methodology to detect MRD through liquid biopsy, research has drastically increased, yielding technological and analytical improvements over recent years [3,7,8]. In this review, we discuss the methodologies that have been developed to detect MRD through liquid biopsy in non-hematological cancers, as well as the potential impact on future trials and clinical practice.

### 1.2. Liquid Biopsy

Liquid biopsy is defined as ‘a test done on a sample of blood to look for cancer cells from a tumor that are circulating in the blood, or for pieces of DNA from tumor cells that are in the blood’ [9]. It was first mentioned by Pantel and Alix-Panabières [10] to describe the use of a blood test to assess the presence and characteristics of a solid tumor. More generally, liquid biopsy refers to all biomarkers that can be detected in the blood: Circulating deoxyribonucleic acid (DNA), proteins such as carcinoembryonic antigen (CEA) or cancer antigen 15.3 (CA15.3), tumor-associated cells, different types of ribonucleic acids (RNAs), and exosomes, among others. In this review, we focus on liquid biopsy performed specifically on blood samples but note that liquid biopsy also includes analyses of other body fluids such as cerebrospinal fluid (CSF), saliva, urine, etc.

Liquid biopsy has several advantages over tumor biopsy. First, it is usually less invasive, dangerous, and painful. Second, it may provide a more comprehensive molecular overview of tumor heterogeneity, possibly reflecting the characteristics of different cancer locations in a particular patient. This cannot be achieved by a single tumor biopsy [11]. Finally, it is easier to obtain repeated blood samples over time to understand the dynamics of response to a specific treatment.

We will briefly review the main advances in the field of liquid biopsy over past years, focusing on the detection of plasma circulating-tumor DNA (ctDNA), one of the most advanced and frequently investigated technologies used to detect MRD [6].

#### 1.2.1. Circulating Nucleic Acids

ctDNA

ctDNA is characterized by tumor-specific genomic alterations. The presence of ctDNA in the blood was first assessed in 1948 [12] and is a proportion of circulating free DNA (cfDNA). Every cell is able to shed DNA into the bloodstream through necrosis, apoptosis, or active secretion [3]. Although cfDNA biology is not fully understood, the quantity of cfDNA depends on several parameters: it can be increased during pregnancy, in case of infection or cell death (i.e., crush syndrome or major physical exercise), or after organ transplantation [13]. In healthy individuals, most cfDNA is derived from hematopoietic cells [14] and is fragmented to a mean length of 166 bp [15].

In the 1970s, Leon et al. showed that the quantity of cfDNA was greater in cancer patients [16]. Tumor cells have the same intrinsic capacity as normal cells to shed DNA, albeit with a higher probability due to higher cellular turnover. DNA with tumor-specific alterations can thus be detected in blood [17]. The DNA shed by tumor cells is more fragmented than that of healthy cfDNA, with the mean length of ctDNA molecules being around 143 bp [15]. The proportion of ctDNA is highly variable and can range from 0.1% to over 10% of the total cfDNA depending on the type of tumor, its size and localization, and tumor staging [18]. For example, ctDNA is detected in more patients with colon adenocarcinoma than in those with glioblastoma, and the amount of ctDNA is higher in the case of metastatic spread compared to localized disease [18]. The methodology of MRD detection through ctDNA and the results of current clinical trials will be discussed in detail in a later paragraph.

Mitochondrial ctDNA

Mitochondrial circulating tumor DNA (mtDNA) is also shed by tumor cells and could potentially be a biomarker of cancer. The quantity of mtDNA may be elevated in some cancers, although not in all [19]. Until now, MRD detection through mtDNA mutations has not been extensively investigated in solid tumors, but promising results have been reported in leukemia patients [20].

Methylation patterns

Epigenetic modification could be divided into three main categories: methylation abnormalities, histone modification, and non-coding RNAs.

DNA methylation is a very powerful mechanism to regulate gene expression [21]: a methyl-group is added to the cytosine on CpG sites by proteins belonging to the family of DNA methyltransferases (DNMT). DNMT3a and DNMT3b are mostly involved in de novo methylation and thus in embryonic differentiation. DNMT1 is responsible for the maintenance of methylation during cell division. In normal cells, methylation occurs in a pattern specific to each cell type and does not occur in promotor regions, as they contain regulatory elements. In cancer cells, methylation patterns are disrupted, with hyper-methylation of promotor regions of tumor suppressor genes and hypo-methylation of promotor regions of oncogenes [21]. This leads to a down-regulation of the expression of tumor suppressor genes and an increased expression of oncogenes. Colon cancer is a seminal example of the role of methylation in tumorigenesis. The methylation of the *MLH1* gene promotor leads to the repression of MLH1 protein expression, resulting in the inactivation of the mismatch repair machinery [22]. Several clinical trials are ongoing with therapies targeted at the mis-methylation process that occurs in cancer cells [23].

Detection of cancer-associated methylation patterns in the blood after surgery in Stage II–III colon cancer was found to be an independent predictive marker of high risk of relapse [24]. However, methylation pattern-based MRD detection did not show high specificity in this study, with 62% of negative patients eventually relapsing after completion of treatment. Detection of methylated-ctDNA (met-ctDNA) may also be a potential biomarker for MRD in breast cancer [25]: the absence of met-ctDNA decrease during neoadjuvant chemotherapy and the presence of met-ctDNA after treatment was correlated with a higher risk of relapse. One concern is that the proportion of patients in whom met-ctDNA was detected prior to treatment was low (20%). Sensitivity could be increased by looking for more than one methylation pattern [26,27]. In these studies, a multiple-gene methylation pattern correlated with treatment response and was associated with worse progression-free survival. Met-ctDNA detection could thus be a powerful biomarker of MRD, but detection methods are not yet sensitive enough to be implemented in clinical practice [28]. Studies are currently ongoing to assess for methylation patterns in blood as a cancer detection tool (NCT04814407, NCT04511559) but also for to monitor disease and determine the presence of MRD (NCT03634826, NCT03737539). The detection of met-ctDNA might be a more powerful tool if combined with other biomarkers.

Non-coding RNA

RNA is composed of a single strand of ribonucleic acids and can be divided into coding RNAs, also called messenger RNA (mRNA), and non-coding RNA (ncRNA). ncRNA is a family of RNA molecules defined by their function and length. ncRNA is subdivided into housekeeping RNA, which includes transfer-RNA and ribosomal-RNA, and regulatory-ncRNA. Regulatory-ncRNA is also subdivided into multiple categories based on structure: long-ncRNA (>200 nucleotides), circular-RNA, and functional ncRNA including micro-RNA, small-ncRNA, small nuclear-ncRNA, and Piwi-interacting-RNA (piRNA) [29]. As these types of RNA are found in the nucleus and cytoplasm of cells, they can also be detected in the bloodstream or in other body fluids, via the same mechanisms as ctDNA [30]. The most investigated are miRNA and lncRNA. miRNAs modulate the expression of genes by binding to mRNA and silencing translation [31]: cancer cells express specific miRNA signatures that can be detected in tumoral tissue and in the bloodstream [30].

Data have shown that it is possible to detect cancer through the detection of cancer-specific miRNAs in testicular germ cell cancer [32], colon cancer [33], ovarian cancer [34,35], gastric cancers [36], and breast cancer [37]. Multiple studies are currently ongoing to assess the potential of miRNA as an MRD biomarker in several cancers, such as breast cancer (NCT04720508), pancreatic cancer (NCT04406831), prostate cancer (NCT04835454), head and neck cancer (NCT04305366), and many others (source: clinicaltrials.gov, accessed on 26 April 2021).

Long non-coding RNA (lncRNA) has also been investigated as a potential biomarker of tumor cells in many cancers [38,39,40,41,42]. LncRNA can be detected in the bloodstream and correlated with tumor stage and treatment response [43,44,45]. Despite showing good sensitivity (~70–80%), the specificity of lncRNA diagnostic tests is too low (~60%) to be applicable in routine practice [43,46]. Most ongoing clinical studies investigating lncRNA focus on its use as a cancer diagnostic tool (NCT03830619, NCT03469544, and NCT04269746). One ongoing clinical trial in triple-negative breast cancer assigns patients to different treatment strategies based on an assessment of their risk of relapse as determined by lncRNA (NCT02641847). To our knowledge, there are no current trials investigating lncRNA to detect MRD (source: clinicaltrials.gov, accessed on 28 April 2021).

Exosomes

Extracellular vesicles (EVs) have a phospholipid bilayer membrane containing bio-products derived from cells that secrete them via an active process. Exosomes are a subset of EVs released by living cells [47,48], ranging in size from about 40 to 160 nm (median 100nm) [49]. Apart from cytosolic and transmembrane proteins, they contain single and double stranded DNA [50], RNA (mRNA, miRNA, lncRNA, cRNA, etc. [49]), and cytosolic metabolites [48]. Exosomes are key players in inter-cellular communication. Their secretion allows cells to induce genetic, epigenetic, and protein transformation of other cells. This method of communication has been investigated in conditions including pregnancy [51], auto-immune disease [52], and cancer [49,53]. Studies have shown that cancer cells may produce up to 20-fold more exosomes than non-cancerous cells [47,54]. Exosomes play a role in tumorigenesis by allowing cell-to-cell communication to induce, for example, epithelial-to-mesenchymal transition (EMT), invasion and migration of cancerous cells, and a tolerant immune environment [49]. Cancer-derived exosomes can be isolated using cancer-specific proteins expressed at their surface [49]. Cancer-specific DNA and RNA modifications are also found in exosomes [49,55,56]. MRD could therefore theoretically be detected through cancer exosomes [55]; however, standardized isolation and quantification methods are lacking [57]. While clinical trials are ongoing to assess the diagnosis of tumors and the detection of MRD through exosomes, also under investigation is the use of the modulating power of exosomes to deliver systemic therapies to specific cells [58].

#### 1.2.2. Circulating Tumor Cells

Circulating tumor cells (CTCs) were first described in 1869 when Ashworth detected malignant cells in the bloodstream of a metastatic cancer patient [59]. It would be more than a century, however, before CTCs were defined and isolated. By definition, CTCs are tumor cells. They can be isolated from other circulating cells based on tumor-specific alterations (immuno-chemistry, size, or density [59]). The tumor-specific genomic and epigenetic alterations that they carry can also be harnessed to detect CTCs [60], with the advantage that their DNA, in contrast to ctDNA, is not fragmented [61].

Although CTCs are indeed shed into the blood from primary tumors or secondary lesions [62], their presence does not necessarily imply the presence, nor predict the development, of micro-metastases. For a cell to be able to metastasize in a specific organ, it needs to undergo MET (mesenchymal to epithelial transition) in order to breach the vascular endothelium and colonize tissue [63]. CTCs with metastatic potential would have to express specific proteins on their surface in order to be recognized by endothelial cells. The FDA-approved CellSearch^®^ [64] isolation technique is based on the detection of one such cell-surface protein: the epithelial cell antigen EpCAM, on the surface of CTCs. This technique would, however, miss any CTCs without an epithelial phenotype [65].

Notwithstanding their limitations, numerous clinical studies have investigated the prognostic role of CTCs. These studies are either quantitative, counting CTCs in the bloodstream, or qualitative, analyzing CTCs for their genome, transcriptome, and/or proteome. The presence of CTCs after neoadjuvant chemotherapy was found to be associated with worse disease-free survival (DFS) in early breast cancer [66,67], naso-pharyngeal carcinoma [68], and lung cancer [69], as well as a worse overall survival (OS) in melanoma [70,71]. Similarly, the detection of CK-19 mRNA in CTCs after curative intent treatment was found to be associated with worse DFS in early breast cancer [72,73]. No doubt due to the difficulty of immediately processing CTCs and the lack of standardized methods for isolation, there is, to our knowledge, only one interventional trial that evaluated the use of CTCs as a marker of MRD [74]: this randomized trial has shown the potential benefit of detecting CTCs to guide adjuvant treatment in hormone-dependent, HER2-negative metastatic breast cancer. There is to our knowledge no interventional trial based on the detection of CTCs ongoing at the moment (source: clinicaltrials.gov, accessed on 15 May 2021). With the support of the European Liquid Biopsy Society, which is working on defining technical guidelines on the collection, processing, and detection of CTCs, we expect more trials in this field to open in the coming years [75].

#### 1.2.3. Circulating Proteins

Today, the most validated MRD-biomarkers remain cancer-specific antigens such as CA125, CA15.3, carcinogen embryonic antigen (CEA), alpha-feto protein (αFP), and prostate-specific antigen (PSA) [76]. Although these are currently used in daily practice to follow patients throughout the course of their disease, they are not specific or sensitive enough to be used alone as biomarkers for MRD [77,78,79,80,81].

## 2. Methodology to Detect Minimal Residual Disease with ctDNA

ctDNA detection and characterization are the most frequently used and investigated methods to detect MRD, based on the presence of tumor-specific genomic alterations. Some applications have already been implemented in daily clinical practice [82]. Next generation sequencing (NGS) refers to several different techniques of high-throughput nucleotide sequencing (sometimes referred to as massively parallel sequencing), based on the extreme miniaturization, parallelization, automation, and digitalization of polymerase chain reaction (PCR)-based “reading while copying” of DNA [83,84]. These methods allow scientists and clinicians to obtain increasingly large quantities of sequencing data within a few hours [83]. The main limiting factor of these sequencing tools is the error rate, which varies from 1% to 0.01% depending on the brand and type of sequencer [85]. High-confidence detection of variants present at a fraction below 1% and therefore requires sufficient depth of coverage (i.e., the number of sequences that “read” any given nucleotide position) in patient as well as control samples, and the use of bioinformatic analyses, and algorithms that allow for the application of adequate quality control criteria.

Other PCR-based techniques such as droplet digital PCR (ddPCR) allow for the detection of DNA variants up to 0.001% [10], although they are suited to the assessment of limited numbers of pre-determined variants, rather than the discovery or determination of large numbers of different variants.

Below, we describe the sequencing methodologies and techniques used to detect ctDNA in the context of MRD application.

### 2.1. Personalized Methods

Tumor-personalized methodologies to detect ctDNA are methods based on prior knowledge of the genomic landscape of the tumor.

#### 2.1.1. Tumor-Customized Based Panels

One of the most validated methodologies to detect ctDNA is the specific interrogation of plasma-derived cfDNA for genomic alterations previously identified in the corresponding tumor. The tumor biopsy or surgical sample is sequenced (e.g., whole exome sequencing (WES) or a large gene panel) to identify somatic mutations (absent, or present at significantly lower fractions in non-tumoral tissue). A custom-made panel is designed accordingly to sequence the altered positions in the plasma. This custom panel, specific to the patient’s tumor, can employ PCR-amplification using specific primers, or hybridization-based capture using specific probes, in order to enrich the regions of interest for NGS [86,87], with a barcoding method typically incorporated to allow for sample multiplexing.

This design has been implemented in several clinical studies in the context of MRD detection [87,88,89,90] and has shown interesting results. The techniques used in these studies, and their main conclusions, are detailed in Table 1. The main advantage of this methodology is the comprehensive view of the genomic landscape of the tumor that is achieved. This provides multiple targets for follow-up in the plasma, thereby optimizing the sensitivity and robustness of detection. By decreasing target size (i.e., the number of nucleotides to be sequenced) as compared to WES or a large gene panel, use of a patient-specific panel can exploit the considerable capacity of NGS sequencers to increase the depth of sequencing. This can allow for the detection of variants present in very low quantities (for example, variant allele frequency (VAF) < 0.01%, based on a high pretest probability of detecting the variant), extremely useful in detecting MRD. Some technologies that combine deep sequencing with unique molecular identifier (UMI) barcoding and bioinformatic pipelines such as Safe-Seq [90] or TARDIS [89] have been shown to detect tumor variant allele frequency as low as 0.002%. However, the choice of which tumor variants will make up the tumor-personalized panel may be subjective or arbitrary, often based on disease-pertinence bio-informatic algorithms, and can omit driver mutations while selecting passenger mutations. The threshold for positivity of ctDNA is also a subjective decision with some authors setting the limit at one mutation per sample [86], while others require the detection of at least two mutations per sample [88].

This technique requires a tumor sample, and this itself creates several issues. First, obtaining a good-quality biopsy can be challenging and sometimes carries risks (bleeding and pain) for the patient depending on the location (e.g., close to blood vessels). Second, patient-specific panels based on sequences from a particular tumor sample fail to capture genomic alterations in other parts of the same tumor, as well as those in distant metastases, due to tumor heterogeneity. Third, the genomic landscape of the primary tumor and its metastases can change over time as treatment and the natural course of the disease select specific tumor clones. The use of a tumor-guided panel therefore carries the risk of missing (potentially informative) alterations in the plasma.

Despite these drawbacks, this method has become the gold standard for detecting ctDNA for MRD due to its ability to detect tumor variants even when present in very low quantities [95].

#### 2.1.2. Custom-Based PCR Assay

Droplet Digital PCR (ddPCR) is a highly specific technique that allows for the detection of individual pre-defined genomic variants, down to a VAF of 0.01%, with promising results [93]. ddPCR has the benefit of giving quantitative results and, unlike quantitative PCR, also presents a proportion of variant allele frequency within the same assay. It also offers a high-sensitivity compared to other techniques, as very low quantity of DNA (~1 ng) can be used to detect a variant [96].

ddPCR probes are designed to detect genomic alterations already identified in the tumor; this method of MRD detection therefore shares many of the same advantages and disadvantages as the custom-based NGS panel technology described above. The use of ddPCR to detect MRD has been demonstrated in hematological neoplasia, when the carcinogenesis is due to a specific mutation (for example BRAF V600E in hairy cell leukemia [97]). However, data, although still scarce in solid tumors, show promising results in tumors where hotspot mutations are well-established as driver mutations [98].

PCR has the very useful potential to reliably detect known single alterations such as the presence of an oncogenic virus. Chronic infections by viruses have been described as a cause of several hematologic and non-hematologic neoplasms [99]. The presence of the virus is usually determined by immunohistochemistry [100], immune-fluorometry assay [101], or PCR on the tumor biopsy [100]. It can also be detected either by assaying for circulating viral antigens or antibodies directed against the virus [102] or the presence of the viral genome in the bloodstream [103]. Studies have mainly focused on detecting specific viral oncogenes in the plasma using PCR-derived methods, for example, quantitative PCR (qPCR) [103], digital PCR (dPCR) [104,105], and droplet digital PCR (ddPCR) [106,107]. Detection of the viral genome to assess MRD has shown interesting results, especially in Human Papilloma Virus (HPV)-derived and Epstein–Barr (EBV)-derived cancers. In HPV-related oropharyngeal cancers, Chera and colleagues [105] have demonstrated the prognostic value of detecting HPV-DNA in the plasma of patients after curative-intent treatment. The detection of EBV after curative-intent treatment has in turn also been shown to be predictive of disease relapse in EBV-related nasopharyngeal carcinoma [103,108]. NGS-based methods can also be used for viral detection [109,110], with the added benefit of assessing for multiple subtypes of virus in a single assay. Nevertheless, the sensitivity of this assay in the MRD setting is yet to be determined.

### 2.2. Non-Personalized Methods

To address the potential of ctDNA as a biomarker, many have chosen a non-personalized or agnostic approach, i.e., one that is not guided by the specific genomic alterations present in a particular patient’s tumor. Gene panels are typically the method of choice, due to their potential to detect any mutations in the neoplasm that are located in the targeted (i.e., sequenced) region of its genome, in one go. As we previously discussed, however, NGS is not the most sensitive method to detect variants present at very low frequency [111,112], necessitating the use of supplementary methods to improve accuracy. For example, to separate real tumor variants from sequencing errors, the use of molecular barcodes (often termed UMIs, for unique molecular identifiers) has been implemented, to lower the error rate of variant-calling and increase the accuracy of VAF estimation [113]. Statistical and bio-informatic tools are used to further separate very low-frequency variants from possible sequencing errors [114].

To detect MRD, two main approaches have been studied: the use of (i) NGS gene panels, either comprehensive or specific to one tumor type, and (ii) PCR for mutations in very specific tumor subtypes.

A synthetic view of the main studies performed using gene panels can be found in Table 2. As shown, it is feasible to use a large panel with over 1000 genes [115], but sophisticated bio-informatic tools and barcoding are essential to detect tumor variants with high confidence. One of the most widely-used strategies is cancer personalized profiling by deep sequencing (CAPP-Seq) [116]. It combines the deep sequencing of several genes (usually >300) implicated in carcinogenesis by the amplification of the target regions, the use of molecular barcoding, and the integrated digital error suppression (iDES) bio-informatic tool [117], specifically designed to detect ctDNA at a very low allele frequency. This technique has shown very promising results when it comes to detecting MRD in specific cancer types. Even without prior knowledge of the genomic mutations in the tumor, this method can detect ctDNA before treatment in >90% of localized NSCLC [116,118]. It is, however, less effective in localized esophageal cancer, detecting ctDNA pre-treatment in only 60% of patients [119]. This may be explained by differences in the genomic landscape and the fact that esophageal cancer seems to shed less tumor DNA than NSCLC [119]. In the post-curative treatment setting, the success rate of the technique drops. Although it shows a good positive predictive value, the negative predictive value still has room for improvement: when ctDNA is detected post-treatment, patients are significantly more likely to experience relapse in the following months [87,120,121]. However, due mainly to technical limitations, not detecting ctDNA after curative-intent treatment does not predict the tumor’s definitive elimination (cf/Table 2). Despite these drawbacks, this technology provides a much more comprehensive view of genomic heterogeneity than tumor-customized methods. It also enhances the understanding of the evolution of the disease as it can detect the emergence of certain clones. Based on these observations, several companies have developed either pan-cancer or cancer-specific panels (Roche Avenio ctDNA targeted kit, Foundation One^®^ LiquidCDx, Guardant360^®^ [122]) to implement this technology in clinical practice. These commercial panels were designed for diagnostic purposes but their use in the detection of MRD still needs to be proven. Other studies have used large panels, with or without UMIs, with interesting results [115,120,123,124]; however, the detection rate of ctDNA undeniably decreases as panel size increases (Table 2).

To increase assay sensitivity in particular populations of interest, several groups have designed smaller gene panels, restricting target size in order to divert data capacity towards maximizing sequencing depth, with promising results in disease detection by ctDNA. Reducing panel size increases coverage depth, as the coverage depth depends on the capacity of the sequencing machine to generate a defined quantity of data per run, the number of samples per run, and the target size, i.e., the number of bases to be sequenced. By reducing the panel-target size, coverage depth increases, and tumor variants may be detected at a much lower VAF. For example, Mes and colleagues detected ctDNA in 67% of patients with head and neck squamous cell carcinomas using a 12-gene panel [127]. For MRD detection, however, no studies to date have used a panel of fewer than 127 genes [123].

PCR techniques may also be used in a non-personalized setting, but their use is limited to specific known, recurrent genomic alterations and clinical situations. For example, the only FDA-approved PCR test, cobas^®^ EGFR mutation test v2 (Roche), detects the presence of 42 *EGFR* mutations, including the hotspot T790M mutation, in patients with NSCLC previously treated with tyrosine kinase inhibitors [128]. This methodology can thus only be used in a specific group of patients and is not a suitable option for most cancers.

### 2.3. Other Methods

Carcinogenesis can be driven by focal mutations (point mutations, small insertions/deletions) in the nucleotide sequence and/or by copy number changes in genes or groups of genes [129]. Tumor copy number alterations (CNA) can be detected by different types of sequencing: regular or shallow WGS [130,131], deep WES [132], or panel sequencing [133]. Detection of CNA in the bloodstream has been used for many years in the prenatal diagnosis of fetal abnormalities [134]. Although several bioinformatic tools have been designed to detect CNA in ctDNA [135,136,137], a consensus on methodology is still lacking. Furthermore, the detection of tumor CNA perhaps does not faithfully reflect the persistence of tumor cells, having been found in treated breast cancer patients without evidence of relapse [138]. One hypothesis that has been tested in squamous cell carcinoma of the head and neck (SCCHN) is that the combined assessment of mutations and CNA [127] may improve ctDNA detection. This has not, however, been tested for MRD detection.

## 3. Clinical Impact of Detecting MRD with ctDNA and Perspectives

As discussed previously, clinicians wish to detect MRD for two main reasons: (i) to offer treatment escalation when MRD is detected after curative-intent therapy with the hope of improving the cancer outcome and (ii) to de-escalate treatment when MRD is not detected with the goal of decreasing treatment toxicity. Therefore, accurate MRD assessment is likely to play an important role in cancer treatment personalization.

Multiple studies in various cancer types (Table 1 and Table 2) have demonstrated a satisfactory positive predictive value of ctDNA detection, as the presence of MRD identified through ctDNA is associated with worse DFS. Nevertheless, several hurdles maintain the negative predictive value of the technique at a low level: detection methods are not sensitive enough to detect ctDNA at very low proportions, new clones not captured by the selected technique can emerge, and the timing of post-treatment sampling can be inappropriate.

### 3.1. Issues

Detecting MRD with ctDNA is still a complex issue as there are no standardized methods ready for implementation in daily practice.

The technical issues have been discussed above. While personalized methods use the most sensitive technologies and can accurately detect tumor variants, sequencing the tumor and then designing patient-specific NGS or PCR assays is costly and time-consuming. Non-personalized methods would therefore theoretically be more practical in clinical practice, as one technique would serve multiple patients suffering from the same tumor type and potentially even be used across different tumor types. Until recently, however, sequencing techniques have not been sensitive enough even with barcoding methodology. The risk of not correctly identifying MRD could negatively impact a patient’s outcome. Moreover, large panels can yield high numbers of variants, and sorting through which to correctly identify driver variants can be unreliable. To exclude non-pathogenic variants, sequenced reads would ideally be aligned not only to reference genomes but also to the germline DNA of the patient using dedicated pipelines, in addition to excluding variants that are frequent in germline variant databases. This would require germline DNA sequencing for all patients.

Clonal hematopoiesis of indeterminate potential (CHIP) [139] might also be a confounding factor—these variants arise from hematopoiesis and might incorrectly be called tumor variants by bioinformatic tools. One way to separate CHIP variants from tumor variants would be to align the sequencing reads on germline DNA collected from circulating white blood cells, such as lymphocytes or peripheral blood mononuclear cells (PBMC), which would also carry the former. Nevertheless, accurately differentiating non-tumor from tumor variants can remain difficult [140,141,142]. Finally, the treatment itself (e.g., radio(chemo)therapy) may induce variants, either in hematopoietic cells (CHIPs) or in cells of the exposed tissue. Separating these variants from true tumor variants is difficult because treatment-induced variants will, like tumor variants, appear at low variant allele frequency and might also occur in genes implicated in carcinogenesis (e.g., *TP53*). Consortia on guidelines to discriminate tumor variants from other variants are emerging [143], and more databases are expanding their pipelines to sort variants, e.g., Varsome Clinical [144]. However, a systematic bioinformatic pipeline has yet to be proven useful in the clinical setting.

The timing of post-treatment sampling is also crucial when trying to adequately detect MRD. ctDNA has a very short half-life (~30 min) [3] and is dependent on active secretion by tumor cells and cell death. During the course of treatment, a decrease in ctDNA has been observed but a precise and universal wash-out time has not been defined [87]. Circulating DNA (cfDNA) can be released into the bloodstream after surgery, mainly due to tissue trauma, and can remain elevated up to four weeks later [145]. Adjuvant therapies usually start 6–8 weeks after primary curative treatment. Assessing for MRD within one to two weeks of curative treatment is, at least for the moment, not technically feasible, and delaying adjuvant therapy until results are available may lower patient prognoses.

Studies are warranted across different cancer types to determine the most effective means of harnessing ctDNA as a biomarker of MRD. Indeed, a patient can be ctDNA-positive several months [146] or just a few weeks before relapse. Some studies have also shown that two positive ctDNA samples collected at two different time points can be more specific than a single positive sample obtained at a unique time point [146].

One of the obstacles to prospective multi-center trials is to perform reproductible assays to detect MRD. Numerous pre-analytical biases exist and include liquid biopsy processing (blood drawing, tubes used, plasma separation, etc.), DNA extraction methodology, sample conservation, equipment used (NGS sequencer, PCR assay), etc.

Despite the issues observed in retrospective studies, prospective trials based on ctDNA as marker of MRD have begun to establish the feasibility of the process.

### 3.2. Current Prospective Trials

Ongoing clinical trials relying on ctDNA as a marker of MRD use four main designs as schematized in Figure 2: prognosis (A), intensification (B), de-escalation (C), and combined (D).

The first, (A), is a non-interventional design that evaluates the prognostic impact of ctDNA detection. Although ctDNA has already been shown to be a prognostic marker in some cancers, trials in tumors where ctDNA has been underexplored, at least until now, are warranted. Such trials are also necessary to evaluate the accuracy of the ctDNA detection assay under investigation. The optimal time to detect ctDNA following curative-intent therapy, and the impact of this timing on prognosis, can also be evaluated in these trials.

The second trial design (B) is an interventional trial that randomizes patients who are ctDNA-positive after receiving standard of care (SOC) treatment ito adjuvant intensification therapy or placebo. This design investigates whether adjuvant treatment can improve OS and/or PFS if ctDNA is detected. Although this trial design is elegant and can potentially improve survival rates, in the event of a negative outcome, it can be difficult to evaluate whether it is the patient selection that is ineffective (ctDNA may not be a good marker to select those in need of adjuvant treatment) or the adjuvant treatment.

The third trial design (C) is the de-escalation of SOC adjuvant treatment. In some tumor types, adjuvant treatment is routinely administered following curative-intent treatment (surgery), but adjuvant treatment may not be required for ctDNA-negative patients. If these trials show non-inferiority in terms of survival between cohorts, they might redefine SOC adjuvant treatment guidelines and spare a subset of patients the burden of additional treatment.

Finally, it is possible to combine these trial designs (D) so that the prognostic impact of ctDNA detection and the added value of adjuvant treatment in the event of post-treatment ctDNA positivity can both be evaluated in a single study.

A non-exhaustive list of the ongoing prospective and interventional trials can be found in Table 3. The majority of ongoing trials are still ‘prognosis’ trials designed to prove the value of ctDNA as a biomarker of MRD. However, ‘escalation interventional’ trials are being launched. One of the first trials to explore these issues is the Australian DYNAMIC trial (ACTRN/12615000381583), in which adjuvant chemotherapy is administered to MRD-positive Stage II colorectal cancer patients after surgery. The COBRA trial (NCT04068103) has almost the same design in Stage II colon cancer and is recruiting at the time of writing. Other ongoing randomized trials are the IM-VIGOR 011 trial (NCT04660344) in invasive urothelial cancer and the Mermaid-1 trial (NCT04385368) in NSCLC, with immunotherapy as adjuvant treatment. The c-TRAK-TN clinical trial (NCT/03145961) in early-stage triple negative breast cancers, currently recruiting in the UK, is a non-randomized trial where adjuvant therapy is given to MRD-positive patients outside SOC. There is no doubt that more trials based on these models will soon be launched. De-escalation trials may soon become available but require careful design due to the risk to patient outcomes.

## 4. Conclusions

The detection of minimal residual disease (MRD) through circulating tumor DNA (ctDNA) is an important potential application of liquid biopsy that could help personalize a patient’s adjuvant treatment. However, several technical issues need to be resolved before these assessments can be implemented in clinical practice. We now need to perform carefully designed randomized interventional clinical trials specifically powered to evaluate if treatment escalation improves cancer outcome in MRD-positive patients and if treatment de-escalation is safe in MRD-negative patients. These future trials will determine the positioning of liquid biopsy in clinical guidelines. For some tumors, there is no doubt that ctDNA will, in the coming years, become a regular biomarker in daily practice.

## Figures and Tables

**Figure 1 cancers-13-05364-f001:**
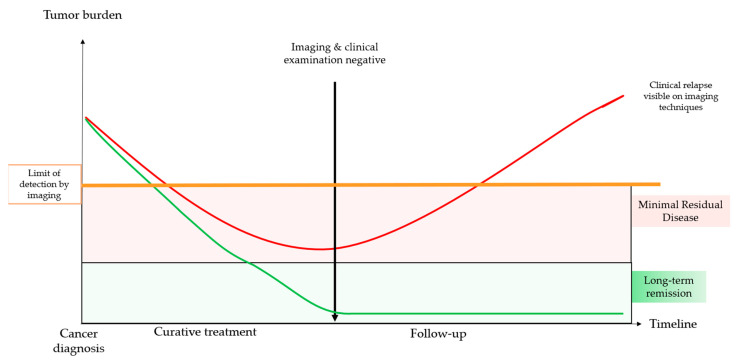
Schematic view of minimal residual disease (MRD).

**Figure 2 cancers-13-05364-f002:**
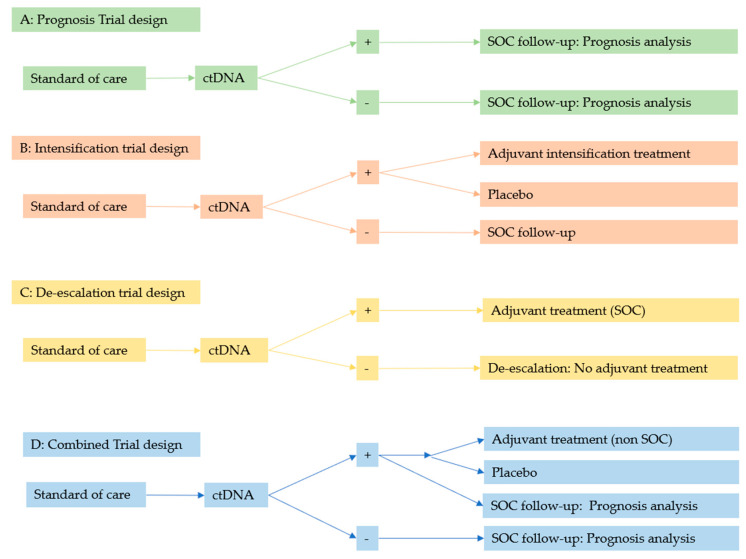
Trial designs based on ctDNA detection. SOC: Standard of care.

**Table 1 cancers-13-05364-t001:** Non-exhaustive list of published trials with personalized methods to identify MRD through ctDNA detection.

**Reference**	**Number of Patients Included (*n*)**	**Tumor Type and Indication**	**Methodology**	**Conclusions**
Early detection of metastatic relapse and monitoring of therapeutic efficacy by ultra-deep sequencing of plasma cell-free DNA in patientswith urothelial bladder carcinoma [91]	68	Muscle invasive bladder cancer treated with neoadjuvant chemotherapy before cystectomy	Tumor sequencing: WES Plasma sequencing: 16 mutations/patient by multiplex PCR.	A total of 76% of ctDNA-positive patients post cystectomy had recurrence (median 96 days before). A total of 0% of ctDNA-negative had recurrence.
Mutation tracking in circulating tumor DNA predicts relapse in early breast cancer [86]	55	Early breast cancer patients receiving neoadjuvant chemotherapy	Tumor sequencing: NGS on panel with 14 known breast cancer driver genes (26). Plasma sequencing: 1 (or more) mutation(s) was (were) followed using ddPCR.	ctDNA was detected in the single post-operative blood test in 19% (7 of 37) of patients. ctDNA detection was predictive of early relapse (median 6.5 months).
Personalized circulating tumor DNA analysis to detect residual disease after neoadjuvant therapy in breast cancer [89]	33	Stage I to Stage III breast cancer	Tumor sequencing: WES Plasma sequencing: Using TARDIS (combinaison of NGS + PCR + UMIs): 6 to 115 mutations per patient.	Before treatment, ctDNA detected in 32 of 32 patients at tumor fractions of 0.002% to 1.06%. Plasma samples after completion of NAT were analyzed in 22 patients. ctDNA+ in 17 out of 22 patients, including 12 out of 13 patients with invasive or in situ residual disease and 5 out of 9 patients with pathological CR. In patients who achieved pathological CR, the median decrease in ctDNA was 96%, whereas in patients with residual disease observed at surgery, the median decrease was 77%.
Targeted next-generation sequencing of circulating-tumor DNA for tracking minimal residual disease in localized colon cancer [92]	94	Resectable colon cancers with plasma available	Tumor sequencing: NGS on custom targeted panel of 29 genes. Plasma sequencing: personalized ddPCR assays for each somatic mutation identified in the tissue.	ctDNA was detected in 63.8% at baseline. ctDNA was detected at 6–8 weeks post-surgery, before starting adjuvant chemotherapy, in 20.3% (14 of 69) patients with plasma available at this time. In ctDNA-positive post-op: 57.1% (8 of 14 patients) experienced reccurence. The presence of ctDNA immediately after surgery was associated with poorer DFS.
Circulating tumor DNA analyses as markers of recurrence risk and benefit of adjuvant therapy for Stage III colon cancer [90]	96	Stage III colon cancer	Tumor sequencing: NGS on 15 genes recurrently mutated in colorectal cancer. Plasma sequencing: 1 mutation/patient with Safe-Seq (NGS + UMIs).	A tumor-specific mutation was detected (ctDNA-positive finding) in the post-surgical plasma sample of 20 of 96 patients (21%). ctDNA was detectable in 15 of 88 (17%) post-chemotherapy samples. Post-surgical ctDNA was detectable in 10 of 24 patients (42%) with recurrence.
Circulating tumor DNA in neoadjuvant-treated breast cancer reflects response andsurvival [88]	84	High-risk earlybreast cancer patients with NAT (I-SPY2 Trial)	Tumor sequencing: WES Plasma sequencing: 16 mutations/patient by multiplex PCR	After NAC, all patients who achieved pCR were ctDNA-negative (*n* = 17, 100%). For those who did not achieve pCR (*n* = 43), ctDNA-positive patients (14%) had significantly increased risk of metastatic recurrence (HR 10.4; 95% CI, 2.3–46.6). Patients who did not achieve pCR but were ctDNA negative (86%) had a similar outcome to those who achieved pCR.
Circulating Tumor DNA predicts pathologic and clinical outcomes following neoadjuvantchemoradiation and surgery for patients with locally advanced rectal cancer [93]	29	Locally advanced rectal cancer	Tumor sequencing: WES Plasma sequencing: personalized ddPCR assays for each somatic mutation identified in the tissue.	Patients with detectable postoperative ctDNA experienced poorer RFS (hazard ratio, 11.56; *p* = 0.007). All patients (4 out of 4) with detectable postoperative ctDNA recurred (positive predictive value = 100%), whereas only 2 out of 15 patients with undetectable ctDNA recurred (negative predictive value = 87%).
Galaxy Study: Preoperative ctDNA levels are detectable in the majority of patients with resectable colorectal cancer [94]	808	Resectable CRC	Tumor sequencing: WES Plasma sequencing: personalized ddPCR assays for each somatic mutation identified in the tissue.	Longitudinal ctDNA positivity at postoperative weeks 4, 12, and 24 was significantly associated with inferior disease-free survival (DFS) with a hazard ratio (HR) of 46.8. Sensitivity of relapse detection was 93.1%. Positivity at postoperative week 4 was significantly associated with inferior DFS with HR 19.5 overall, and HR 24.4 in pathologic Stage I–III, indicating it is a suitable time point for ctDNA-based adjuvant study.
Dynamics of cell-free tumour DNA correlate with treatment response of head and neck cancer patients receiving radiochemotherapy [87]	20	Non-resecable locally advanced head and neck squamous cell carcinoma	Tumor sequencing: NGS with 327 genes panel. Plasma sequencing: 127 driver mutations + E7 NGS panel	Baseline: ctDNA-positive: 17/20 patients Post RCT ctDNA-positive-: 2/16 patients Eight patients relapsed: 2ctDNA-positive Eight patients without relapse: 8ctDNA-negative PPV 100%, Sn 25%

WES: whole exome sequencing; PCR: polymerase chain reaction; ddPCR: droplet digital polymerase chain reaction; NGS: next generation sequencing, UMI: unique molecular identifier; NAT: neoadjuvant treatment; CR: complete response; NAC: neoadjuvant chemotherapy; pCR: pathological complete response; DFS: disease-free survival; RFS: relapse-free survival; HR: hazard ratio; RCT: radio-chemotherapy; PPV: positive predictive value; Sn: sensitivity; ctDNA: circulating tumor DNA.

**Table 2 cancers-13-05364-t002:** Non-exhaustive list of published trials with non-personalized methods to identify MRD through ctDNA detection.

**Reference**	**Number of Patients (** ** *n* ** **)**	**Tumor Type and Indication**	**Methodology**	**Conclusions**
Early detection of molecular residual disease in localized lung cancer by circulating tumor DNA profiling [118]	40 54 healthy controls	Curative intent for Stage I–III lung cancer	Plasma sequencing: CAPP-Seq 128 genes most frequently mutated in lung cancer.	94% of patients with MRD were ctDNA-positive in post-treatment plasma samples. Patients were ctDNA-positive before radiological relapse (72%) (5.2months). 53% of ctDNA-positive patients had actionable targets.
Circulating tumor DNA analysis for detection of minimal residual disease after chemoradiotherapy for localized esophagealcancer [119]	45	Stage IA to Stage IIIB esophageal cancers (adenocarinoma or squamous cell carinoma)	Plasma sequencing: CAPP-Seq Esophageal specific panel	Baseline ctDNA-positive: 27/45 (60%). Post CRT ctDNA-positive: 5/31 (16%). Patients with detectable ctDNA post-CRT also had significantly increased risk of disease progression (HR 18.7, *p* < 0.0001), distant metastasis (HR 32.1, *p* < 0.0001), and disease specific death (HR 23.1, *p* < 0.0001).
Post-radiation circulating tumor DNA as a prognostic factor in locally advanced esophageal squamous cell carcinoma [120]	25	Resectable esophageal squamous cell carcinoma	Plasma sequencing: NGS on a custom designed 180 genes panel	At baseline, 100% ctDNA-positive. Post radiotherapy: 14/24 (58%) ctDNA-positive 10/24 (42%) ctDNA-negative In the 14 ctDNA-positive patients, 11 patients had a documented follow-up: 90.9% (10/11) had documented disease recurrence. In the 10 ctDNA-negative patients, 8 patients had documented follow-up: 50% (4/8) had documented disease recurrence. Patients who were ctDNA-positive exhibited a marginally significant reduction in PFS (*p* = 0.047) and a significantly decreased OS (*p* = 0.005) compared to patients who were ctDNA-negative.
Minimal residual disease detection using a plasma-only circulating tumor DNA assay in colorectal cancer patients [121]	84	Resectable colorectal cancer	Plasma sequencing: Guardant Reveal™ test using NGS custom based panel for the detection of somatic and epigenitic abberations.	Fifteen patients had detectable ctDNA and all 15 recurred. Of 49 patients without detectable ctDNA at the landmark timepoint, 12 (24.5%) recurred. Landmark recurrence sensitivity and specificity were 55.6% and 100%. Integrating epigenomic signatures increased sensitivity by 25–36% versus genomic alterations alone.
Prognostic implications of preoperative versus postoperative circulating tumor DNA in surgically resected lung cancer patients: a pilot study [125]	20	Stage IIA–IIIA lung cancer	Plasma sequencing: CAPP-Seq on a commercial 197 genes panel (Roche Diagnostics).	Eight patients (40%) were positive for preoperative ctDNA. Four patients (20%) were positive for postoperative ctDNA, and this was significantly correlated with histological grade (3 vs. 1 or 2, *p* = 0.032). Postoperative positivity for ctDNA also predicted shorter recurrence-free survival (RFS).
Circulating tumor DNA as a prognostic biomarker in localized non-small cell lung cancer [123]	77	Resectable NSCLC	Plasma sequencing: NGS (cSMART assay) on a custom 127 gene panel	Postoperative ctDNA-positive patients also associated with a lower RFS (HR = 3.076, *p* = 0.0015) and OS (HR = 3.195, *p* = 0.0053). Disease recurrence occurred among 63.3% (19/30) of postoperative ctDNA-positive patients. Most of these patients 89.5% (17/19) had detectable ctDNA within 2 weeks after surgery.
Circulating tumor DNA as a potential marker to detect minimal residual disease and predict recurrence in pancreatic cancer [115]	27	Operable pancreatic cancer	Plasma sequencing: NGS on a large (1.017) gene panel	ctDNA was detected in 18 of 27 preoperative plasma samples, resulting in a detectable rate of 66.67%. Seven days after surgical resection, the status of ctDNA changed in 19 patients. Of these, one turned positive and 10 became completely negative. Patients who were ctDNA-positive postoperatively had a markedly reduced disease-free survival (DFS) compared to those who were ctDNA-negative. A positive postoperative ctDNA status was an independent prognostic factor for DFS.
Deep sequencing of circulating tumor DNA detects molecular residual disease and predicts recurrence in gastric cancer [124]	46	Stage I–III gastric cancer	Plasma sequencing: NGS with Enrich Rare Mutation Sequencing (ER-Seq) assay on a custom driver mutation panel	ctDNA was detected in 45% of treatment-naïve plasma samples. All patients with detectable ctDNA in the immediate post-operative period eventually experienced recurrence. Post-operative samples (collected prior to any adjuvant chemotherapy; 9–48 days after surgery) showed that ctDNA was detected in 18% (7 out of 38) of evaluable patients. ctDNA positivity after surgery was strongly associated with an increased risk of relapse (100% recurrence in the positive group vs. 32% in the negative group), worse DFS (*p* < 0.0001), and worse OS (*p* = 0.0007).
Circulating tumor DNA analyses as a potential marker of recurrence and effectiveness of adjuvant chemotherapy for resected non-small cell lung cancer [126]	38	Resectable NSCLC	Plasma sequencing: NGS on a custom 425 genes panel	Preoperative plasma samples, ctDNA+ in 19 (50%) patients ctDNA was detected post-chemotherapy in 8 out of 36 (22.2%) patients and was associated with an inferior RFS (HR, 8.76; *p* < 0.001).

NGS: next generation sequencing; NSCLC: non-small cell lung cancer; PFS: progression-free survival; OS: overall survival; DFS: disease-free survival; RFS: relapse-free survival; HR: hazard ratio; CAPP-Seq: cancer personalized profiling by deep se-quencing.

**Table 3 cancers-13-05364-t003:** Non-exhaustive list of ongoing trials based on ctDNA detection.

Name of Trial	NCT	Tumor Type	Primary Endpoint	Type of Trial
circTeloDIAG: liquid biopsy in glioma tumor	NCT04931732	Glioma	Sensitivity and specificity of the circTeloDIAG assay at the time of surgery	A: Prognosis trial design
Liquid biopsy in head and neck cancer	NCT099326468	HNSCC	Compare liquid biopsy to PET-CT to evaluate MRD	A: Prognosis trial design
LIQUID	NCT049443406	Gastric cancer	Evaluate the prognosis role of liquid biopsy in locally advanced gastric cancer	A: Prognosis trial design
NSCLC heterogeneity in early-stage patients and prediction of relapse using a personalized “liquid biopsy”	NCT03771404	NSCLC	Correlate the liquid biopsy information to disease recurrence	A: Prognosis trial design
T-MENC	NCT03838588	NSCLC	The concordance of the plasma ctDNA detection status with PFS and OS after radical resection or/and under adjuvant treatment	A: Prognosis trial design
PEGASUS trial	NCT04259944	Colon cancer	Proving the feasibility of using liquid biopsy to guide post-surgical and post-adjuvant clinical management in MSS Stage III and Stage II T4N0 colon cancer	C: De-escalation trial design with several arms depending on de-escalation regime
HCCGenePanel	NCT04111029	Hepatocarcinoma	Prove response to locoregional therapy	A: Prognosis trial design
Liquid biopsy in monitoring the neoadjuvant chemotherapy and operation in gastric cancer	NCT03957564	Gastric cancer	Explore the clinical value of CTC, ctDNA, and cfDNA in neoadjuvant chemotherapy and operation of resectable or locally advanced gastric cancer	A: Prognosis trial design
PROJECTION	NCT04246203	Pancreatic cancer	Prognostic role of circulating tumor DNA in resectable pancreatic cancer	A: Prognosis trial design
ctDNA Lung RCT	NCT049666663	NSCLC	To evaluate whether the presence of circulating tumor DNA (ctDNA) in the blood can help to predict whether giving adjuvant treatment after surgery can decrease the risk of cancer recurrence.	B: Intensification trial design with several arms
Verification of predictive biomarkers for pancreatic cancer treatment using multicenter liquid biopsy	NCT04241367	Pancreatic cancer	Verification of predictive biomarkers for pancreatic cancer treatment	A: Prognosis trial design
Cell-free tumor DNA in head and neck cancer patients	NCT03942380	Head and neck cancer	Measure the percentage of recurrence in head and neck cancer patients through serial monitoring with liquid biopsy	A: Prognosis trial design
MARTINI	NCT04853420	Solid malignancies	Minimal residual disease: a trial using liquid biopsies in solid malignancies	A: Prognosis trial design
WHENII	NCT03481101	NSCLC	Evaluate early response to chemotherapy in NSCLC	A: Prognosis trial design
PECAN	NCT03540563	HNSCC	ctDNA as a biomarker for treatment response	A: Prognosis trial design
Serial ctDNA monitoring during adjuvant capecitabine in early triple negative breast cancer	NCT04768426	Triple negative breast cancer	Detection levels of ctDNA during adjuvant treatment	A: Prognosis trial design
LiBReCA	NCT03699410	Rectal cancer	Investigate the value of liquid biopsies to predict tumor response after neoadjuvant chemo-radiotherapy in patients with locally advanced rectal cancer	A: Prognosis trial design
Monitoring efficacity of radiotherapy in lung cancer and esophageal cancer	NCT04014465	Lung and esophageal cancer	Clinical value of efficacy evaluation and prognosis of ctDNA detecting technique in patients with radiotherapy	A: Prognosis trial design
MRD monitoring in lung cancer after resection	NCT04976296	Lung cancer	MRD monitoring	A: Prognosis trial design
PRE-MERIDIAN	NCT04599309	Locally advanced head and neck squamous cell carcinoma	Number of high-risk HNSCC with successful ctDNA detection after standard treatment	A: Prognosis trial design
TOMBOLA	NCT04138628	Bladder cancer	Treatment of metastatic bladder cancer at the time of biochemical relapse following radical cystectomy	B: Intervention trial design
Adjuvant durvalumab in early-stage NSCLC patients with ctDNA MRD	NCT04585477	NSCLC	Durvalumab as adjuvant treatment in ctDNA-positive patients	B: Intervention trial design
Study of ctDNA guided change of treatment for refractory MRD in colon adenocarcinoma	NCT04920032	Colon adenocarcinoma	Adjuvant TAS-102 + iritotecan in ctDNA-positive colon cancer patients	B: Intervention trial design
Minimal residual disease assessment in patients with colorectal cancer: MIRDA-C study	NCT04739072	Colorectal cancer	Improve the detection of MRD	A: Prognosis trial design
c-TRAK-TN	NCT03145961	Early-stage triple negative breast cancer	A randomized trial using ctDNA mutation tracking to detect MRD and trigger patient intervention.	B: Intervention trial design
CITCCA	NCT04726800	Colorectal cancer	ctDNA as a prognostic and predictive marker in colorectal cancer	A: Prognosis trial design
Clearance of ctDNA big ten cancer research consortium	NCT04367311	NSCLC	Clearance of ctDNA under adjuvant treatment	A: Prognosis trial design
Personalized escalation of consolidation treatment following chemoradiotherapy and immunotherapy in Stage III NSCLC	NCT04585490	NSCLC	Adjuvant therapy in ctDNA-postive patients	B: Intervention trial design
Measuring MRD in colorectal cancer after primary surgery and resection of metastases	NCT03189576	Colorectal cancer	Measuring MRD	A: Prognosis trial design
IMPROVE-IT	NCT03748680	Colorectal cancer	Implementing non-invasive ctDNA analysis to optimize the operative and post-operative treatment of colorectal cancer	B: Intervention trial design
DYNAMIC-III	ACTRN/12615000381583	Colon cancer	Adjuvant therapy in ctDNA-positive patients	B: Intervention trial design
COBRA	NCT04068103	Colon cancer	Adjuvant therapy in ctDNA-positive patients	B: Intervention trial design
IM-VIGOR 011	NCT04660344	Bladder cancer	Adjuvant therapy (atezolizumab) in ctDNA-positive patients	B: Intervention trial design
MERMAID-1	NCT04385368	NSCLC	Adjuvant therapy (durvalumab) in ctDNA-positive patients	B: Intervention trial design
BESPOKE Study of ctDNA Guided Therapy in Colorectal Cancer	NCT04264702	Colon cancer	Adjuvant chemotherapy or observation (choice by treating clinician) in ctDNA positive patients	B: Intervention trial design

NSCLC: non-small cell lung cancer; HNSCC: head and neck squamous cell carcinoma; ctDNA: circulating tumor DNA; MRD: minimal residual disease; PFS: progression-free survival; OS: overall survival. This table has been established using the following keywords on ClinicalTrials.gov: “ctDNA”, “Liquid biopsy” with recruitment status: “not yet recruiting”, “recruiting”, “enrolling by invitation”, “active, not recruiting”, and “unknown status”.

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
