# Peer review of "Liquid Biopsy to Detect Minimal Residual Disease: Methodology and Impact"

_cancers, 2021, doi:10.3390/cancers13215364_

Round 1

Reviewer 1 Report

Honorè N. et al wrote an interesting review about the feasibility to introduce liquid biopsies in clinical practice as predictor factor that could notably improve patient outcome and life quality. I think that the manuscript is well written and almost exhaustive starting to a dissection of the different type of liquid biopsies, through the technique advantages and disadvantages and closing the paper with retrospective and prospective list of clinical trials in which liquid biopsy analysis is performed. 

Major issue:

Nevertheless, in my opinion the authors should better define  that this review is prefentially focused on data derived from solid tumors. Actually, minimal residual disease monitoring by liquid biopsy is an open issue also in hematologic fields, with good results and applications expecially in B- lymphoid neoplasia. 

Moreover, the "custom-based PCR assay" chapter should better describe ddPCR advantages, in terms of sensitivity, and its applications to MRD purposes rather than quantitative PCR approached especially in those tumors featured by known mutation as molecular marker. 

In the "Issues" chapter the authors should also address the discussion to pre-analitical bias that could reduce  the multi center trial feasibility.

Overall, in my opinion, the manuscript is good but too long and full of technical and clinical informations. Maybe the authors should decide to focus  their attention and the follow description only to ctDNA and their application in MRD monitoring

Minor issues:

row 128: "repression of the expression" should maybe changed with "down regulation of" or "decreased expression of"

References section:

reference 9-75: please refer as reference 1

reference 91-92-99-129 are they book? it is not clear

Author Response

Reviewer 1: Honorè N. et al wrote an interesting review about the feasibility to introduce liquid biopsies in clinical practice as predictor factor that could notably improve patient outcome and life quality. I think that the manuscript is well written and almost exhaustive starting to a dissection of the different type of liquid biopsies, through the technique advantages and disadvantages and closing the paper with retrospective and prospective list of clinical trials in which liquid biopsy analysis is performed. 

Thank you very much for these very nice comments.

Major issue:

Nevertheless, in my opinion the authors should better define  that this review is prefentially focused on data derived from solid tumors. Actually, minimal residual disease monitoring by liquid biopsy is an open issue also in hematologic fields, with good results and applications expecially in B- lymphoid neoplasia. 

Thank you for highlighting this very important precision. We have added this precision in the abstract, the summary and the article (page 2 lines 46 & 67).  Although, in agreement with Reviewer n°1, minimal residual disease monitoring by liquid biopsy is also a topic of interest in hematological cancers, we have decided to focus this review on solid cancers to avoid producing an article too long.

Moreover, the "custom-based PCR assay" chapter should better describe ddPCR advantages, in terms of sensitivity, and its applications to MRD purposes rather than quantitative PCR approached especially in those tumors featured by known mutation as molecular marker. 

Thank you for this comment.  Indeed to be more precise we have updated the “custom-based PCR assay” paragraph (see below). We hope these added remarks gives a more comprehensive view of the use of ddPCR in the detection of MRD.

"Droplet Digital PCR (ddPCR) is a highly specific technique that allows for the detection of individual pre-defined genomic variants, down to a VAF of 0.01%, with promising results [96]. ddPCR has the benefit of giving quantitative results and, unlike quantitative PCR, also present a proportion of variant allele frequency within the same assay. It also offers a high-sensitivity compared to other techniques, as very low quantity of DNA (~1ng) can be used to detect a variant [97].

 ddPCR probes are designed to detect genomic alterations already identified in the tumor; this method of MRD detection therefore shares many of the same advantages and disadvantages as the custom-based NGS panel technology described above. The use of ddPCR to detect MRD has been demonstrated in hematological neoplasia, when the carcinogenesis is due to a specific mutation (for example BRAF V600E in hairy cell leukemia[98]). However, data, although still scarce in solid tumors, show promising results in tumors where hotspot mutations are well established as driver mutations[99].  "

In the "Issues" chapter the authors should also address the discussion to pre-analitical bias that could reduce  the multi center trial feasibility.

Thank you very much for this very useful comment. We agree that pre-analytical bias should be thought of before multicentered clinical trials. We have added a small paragraph (see below) page 16, line 490 to discuss this topic.

One of the obstacles to prospective multi-center trials is to perform reproductible assay to detect MRD. Numerous pre-analytical bias exists and include liquid biopsy processing (blood drawing, tubes used, plasma separation, etc…), DNA extraction methodology, sample conservation, equipment used (NGS sequencer, PCR assay, …), etc…

Overall, in my opinion, the manuscript is good but too long and full of technical and clinical informations. Maybe the authors should decide to focus  their attention and the follow description only to ctDNA and their application in MRD monitoring

Thank you for this comment. It is not in accordance with the comments of other reviewers, that is why we did not reduce the length of this article. However, we thank reviewer n°1 for these very insightful remarks, and hope the adjustments made will answer the main issues.

Minor issues:

row 128: "repression of the expression" should maybe changed with "down regulation of" or "decreased expression of"

Thank you for this correction: the change has been made.

References section:

reference 9-75: please refer as reference 1

reference 91-92-99-129 are they book? it is not clear

 Thank you for spotting these mistakes that occured during the editing process. Corrections have been made.

Reviewer 2 Report

Its a well written comprehensive review with good picture and table representation of data. Its well written with good subsections and flow of reading is easy. All the best. 

Author Response

Reviewer 2: Its a well written comprehensive review with good picture and table representation of data. Its well written with good subsections and flow of reading is easy. All the best. 

Thank you very much for this nice comment.

Reviewer 3 Report

This is a very comprehensive review of the various methods used in the detection of MRD. In this review, the authors discussed the different methodologies used and investigated to detect MRD, their respective potentials and issues, and the clinical impacts that MRD detection will have on the management of cancer patients. It is very well written and will likely be referenced many times. I have no substantive corrections.

Author Response

Reviewer 3: This is a very comprehensive review of the various methods used in the detection of MRD. In this review, the authors discussed the different methodologies used and investigated to detect MRD, their respective potentials and issues, and the clinical impacts that MRD detection will have on the management of cancer patients. It is very well written and will likely be referenced many times. I have no substantive corrections.

Thank you very much for this nice comment.